# Predictive factors of relapse after dose reduction of oral 5-aminosalicylic acid in patients with ulcerative colitis in the remission phase

Akira Madarame[1☯*], Masakatsu Fukuzawa[1☯], Yoshiya Yamauchi[1‡], Shin Kono[1‡], Akihiko Sugimoto[1‡], Hayato Yamaguchi[1‡], Takashi Morise[1‡], Yohei Koyama[1‡], Kumiko Uchida[1‡], Maya Suguro[1‡], Taisuke Matsumoto[1‡], Kagawa Yasuyuki[1‡], Takashi Kawai[2‡], Takao Itoi[1‡]

1 Department of Gastroenterology and Hepatology, Tokyo Medical University Hospital, Tokyo, Japan,
2 Department of Gastroenterological Endoscopy, Tokyo Medical University Hospital, Tokyo, Japan

☯ These authors contributed equally to this work.
‡ These authors also contributed equally to this work.
* madarame@tokyo-med.ac.jp

**Data Availability Statement:** All relevant data are within the manuscript and its S1 Appendix files.

## Abstract

### Objectives

Useful indices to determine whether to reduce the dose of 5-aminosalicylic acid (5-ASA) in patients with ulcerative colitis (UC) during remission remain unclear. We aimed to analyze the rate and risk factors of relapse after reducing the dose of oral 5-ASA used for maintenance therapy of UC.

### Methods

UC patients whose 5-ASA dose was reduced in clinical remission (partial Mayo score of $\leq$ 1) at our institution from 2012 to 2017 were analyzed. Various clinical variables of patients who relapsed after reducing the dose of oral 5-ASA were compared with those of patients who maintained remission. Risk factors for relapse were assessed by univariate and multivariate logistic regression analyses. Cumulative relapse-free survival rates were calculated using the Kaplan–Meier method.

### Results

A total of 70 UC patients were included; 52 (74.3%) patients maintained remission and 18 (25.7%) patients relapsed during the follow-up period. Multivariate analysis indicated that a history of acute severe UC (ASUC) was an independent predictive factor for clinical relapse ($p$ = 0.024, odds ratio: 21, 95% confidence interval: 1.50–293.2). Based on Kaplan–Meier survival analysis, the cumulative relapse-free survival rate within 52 weeks was 22.2% for patients with a history of ASUC, compared with 82.0% for those without. the log-rank test showed a significant difference in a history of ASUC ($p$ < 0.001).

**Funding:** The authors received no specific funding for this work.

**Competing interests:** The authors have declared that no competing interests exist.

## Conclusions

Dose reduction of 5-ASA should be performed carefully in patients who have a history of ASUC.

## Introduction

Ulcerative colitis (UC) is a chronic inflammatory disease affecting the colon, which is characterized by an unpredictable course, usually involving periods of flare-up and remission. The pathogenesis of UC involves a variety of factors, including genetic predisposition, epithelial barrier defects, dysregulated immune responses, and environmental factors [1]. Although the incidence of UC remains stable in Western countries, it is increasing worldwide [2]. Previous studies have demonstrated that 5-aminosalicylic acid (5-ASA) is effective for UC, regarding both the induction of and maintenance of remission [3, 4], and is the first-line therapy for patients with mild to moderate UC [5]. Since the disease is chronic and intermittently relapses, treatment with 5-ASA should be continued not only when symptoms flare up, but also when symptoms are not present.

As 5-ASA is an expensive drug, its cumulative use is likely to have considerable financial burden on patients and healthcare agencies. However, the existing guidelines state that because of limited evidence, no recommendations can be made on when and in whom a reduction in the dose of 5-ASA can be performed after obtaining clinical remission. In the European Crohn's and Colitis Organization (ECCO) guidelines, there is a statement the effective dose of oral mesalamine to maintain remission is 2 g/day [5]. Several studies have reported the dose-response data of 5-ASA for the maintenance of remission. Giaffer et al. investigated the efficacy of two doses of balsalazide for maintaining remission of UC, and found that the relapse rate at 12 months on 2 g/day balsalazide was higher (55%) than that on 4 g/day (37%) (RR: 0.66; 95% CI: 0.45–0.97) [6]. Kruis et al. compared different doses of Salofalk granules (3 g/day vs 1.5 g /day) and reported that the relapse rate was lower for patients treated with 3 g/day than 1.5 g/ day (RR: 0.65; 95% CI: 0.49–0.86) [7]. On the other hand, other studies showed that there was no significantly difference in the rate of relapse among patients receiving 5-ASA doses of more than 2 g/day. Pica et al. compared patients treated with 4.8 g/day versus 2.4 g/day of Asacol, and found no significant difference in efficacy, with a 29% relapse rate for 4.8 g/day and a 36% relapse rate for 2.4 g/day (RR: 0.80, 95% CI: 0.46–1.38) [8]. Green et al. reported that 32% of patients treated with 6 g/day and 23% of patients treated with 3 g/day of Balsalazide experienced a relapse, demonstrating no significant difference in efficacy [9]. In clinical practice, the decision of whether to continue a high dose of 5-ASA or reduce the dose in UC patients who are in remission is still based on the judgement of individual doctors.

Therefore, in this study, we retrospectively investigated the clinical outcomes of UC patients during the 52 weeks after reducing the dose of 5-ASA, and the clinical characteristics of the patients who relapsed after reducing the 5-ASA dose after achieving remission, at a single center in Japan. The aim of this study was to assess the rate and risk factors of relapse after reducing the dose of oral 5-ASA for maintenance therapy.

## Materials and methods

### Patients

**Population and definition of inclusion criteria.** This retrospective cohort study analyzed consecutive UC patients in remission, who were treated with reduced doses of 5-ASA for at

least some of the period from February 2012 and November 2017, at the Department of Gastroenterology and Hepatology, Tokyo Medical University (Tokyo, Japan). To ensure the inclusion of all patient, electronic medical records and the established database of UC were reviewed. Patients who were prescribed 5-ASA (Pentasa®, the extended-release form of mesalamine; Kyorin Pharmaceutical Co., Ltd., Tokyo, Japan, or Asacol®, the delayed-release form of mesalamine; Zeria Pharmaceutical Co., Ltd., Tokyo, Japan) were extracted. All patients were at least 18 years of age and had an established UC diagnosis based on standard clinical, endoscopic, and historical criteria, according to the criteria of the Japanese Research Committee on Inflammatory Bowel Disease [10].

Patients who had received a high dose of oral 5-ASA for maintenance therapy and who maintained remission for more than 3 months were included in the study. All patients were in clinical remission at the time of starting the reduction of oral 5-ASA, and were also in corticosteroid-free remission for more than 3 months prior to their inclusion in the study. Patients who received combination therapy (topical therapies, immunomodulators, corticosteroids, calcineurin inhibitors, anti- tumor necrosis factor (TNF)-α agents, and granulocyte apheresis) for at least 3 months prior to dose reduction of oral 5-ASA, whose internal medicine adherence was estimated to be less than 80% by interview by the attending physician, whose observation period was less than 52 weeks, and who were pregnant were excluded.

Clinical disease activity was assessed using the partial Mayo (pMayo) score [11]. Clinical remission was defined as pMayo score of $\leq 1$, and clinical relapse was defined as pMayo score of $\geq 3$ [12]. Acute severe UC (ASUC) was defined as the passage of six or more bloody stools per day with the presence of one or more additional Truelove and Witts (TW) criteria (pulse > 90 bpm; temperature > 37.8 ˚C; hemoglobin < 105 g/L; erythrocyte sedimentation rate [ESR] >30 mm/h) [13]. Because ESR is less frequently used than C-reactive protein (CRP) in current practice, CRP > 30 mg/L was used instead of ESR. This marker is often used and has been reported in other studies [14, 15] and in the ECCO consensus [16]. A high dose of 5-ASA was defined as 3,000 to 4,000 mg of Pentasa or 3,600 mg of Asacol. A low dose of 5-ASA was defined as 2,000 mg of Pentasa or 2,400 mg of Asacol. Data were collected on the patients' demographic details (sex and age) and clinical parameters (age at diagnosis, duration of disease at admission, and date of admission). After patient identification, a complete chart review was performed by two gastroenterologists (AM and MF). Any discrepancies were resolved by discussion with the senior author. The flow diagram of this study is shown in Fig 1.

## Endpoints

The primary endpoint of the study was the identification of predictive factors for relapse during the 52 weeks after reducing the dose of oral 5-ASA. Secondary endpoints were the relapse rate until 52 weeks, the cumulative relapse-free survival rates during the follow-up period, and reinduction therapy for the relapse.

## Statistical analysis

Baseline characteristics of the patients were summarized as the mean (standard deviation) or median (range) for continuous variables, and as frequency (%) for categorical variables. Statistical differences in patient characteristics between the patients who relapsed and those who maintained remission were assessed using the Mann-Whitney U test for continuous variables and the chi-squared test (or Fisher exact test) for categorical variables. Logistic regression was used to perform multivariate analysis of risk factors for relapse, including sex, age, age at diagnosis, duration of disease, duration to disease remission, pMayo score, type of 5-ASA, location

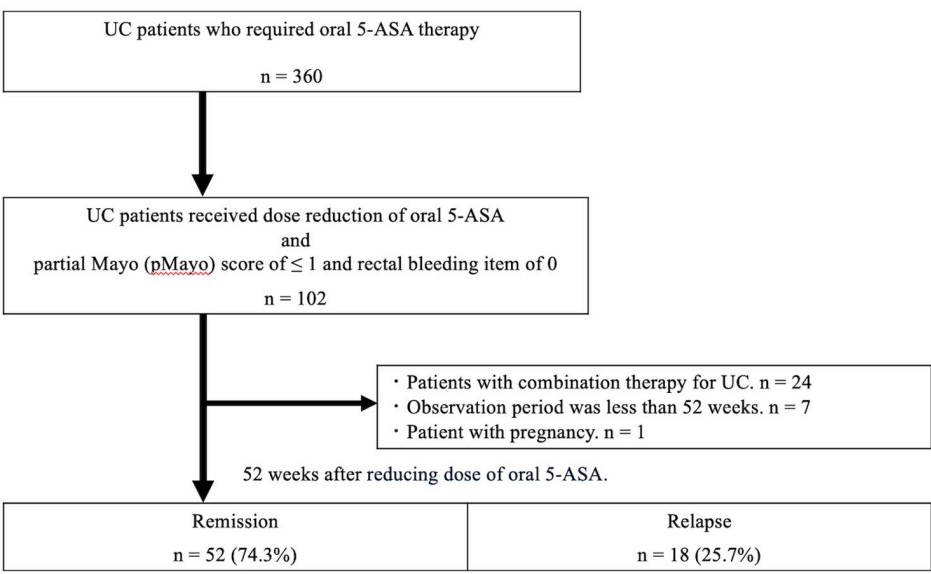

**Fig 1. Flow diagram of this study.**

of disease, history of thiopurine use, history of steroid use, and history of ASUC. At our institution, a CRP level of < 3 mg/L could not be accurately measured in most of the patients. Therefore, CRP was analyzed as a categorical variable for < 3 mg/L and ≥ 3mg/L. In Japan, fecal calprotectin has been available for measurement since June 2017. Therefore, it was almost impossible to measure fecal calprotectin in this study. The cumulative relapse-free survival rates were calculated using the Kaplan–Meier method. All statistical analyses were performed using IBM SPSS Statistics version 26 software (IBM Corp., Armonk, N.Y., USA). Two-sided $p$-values of less than 0.05 were considered to indicate statistically significant differences between the patients who relapsed and those who maintained remission.

## Ethical approval

This study was approved by the Ethics Committee of Tokyo Medical University School of Medicine (study approval no.: T2019-0186), and was conducted in accordance with the principles of the Declaration of Helsinki. Because this study was a retrospective cohort study, the Ethics Committee of our hospital waived the requirement for informed consent. Information about this study was posted in Tokyo Medical University Hospital. Patient records and information were anonymized and de-identified before analysis.

## Results

### Patients and baseline characteristics

Between February 2012 and November 2017, 360 patients at our hospital were treated with oral 5-ASA to maintain clinical remission of UC. There were 102 patients who underwent a dose reduction of 5-ASA and had a pMayo score of ≤ 1 at the start of the observation period. Among them, 24 patients who underwent combination therapy for UC (topical therapies, immunomodulators, corticosteroids, calcineurin inhibitors, anti-TNF agents, and granulocyte apheresis), seven who did not complete follow-up to 52 weeks, and one who was pregnant were excluded from the study, and a total of 70 patients were eligible for inclusion. Table 1 shows the baseline characteristics of the 70 patients fitting the inclusion criteria. The patients

had a median age of 44 (range: 20–81) years, and 61% were men. The median disease duration and median duration of remission were 4.5 (0–37) years and 1 (1–10) year, respectively. Regarding previous use of medication, there were 16 patients with a history of corticosteroid use and three with a history of thiopurine use. The cumulative dose of corticosteroids was available in 14 of 16 patients, with a median of 2555 mg (1120–4970 mg). Two patients received medication of steroids at other hospitals, so details were not available (S1 Appendix). There were no patients with a history of use of anti-TNF agents or calcineurin inhibitors. Eight patients had a history of ASUC. 43 patients were prescribed Pentasa, of which 34 were on 3000 mg/day and 9 were on 4000 mg/day before dose reduction. 27 patients were pre-scribed Asacol, of which all were on 3600 mg/day before dose reduction. Patients prescribed Pentasa were reduced to 2000 mg/day, and patients prescribed Asacol were reduced to 2400 mg/day.

## Risk factors for clinical relapse after dose reduction of 5-ASA

As shown in Table 2, univariate analysis (the Mann-Whitney U test for continuous variables and the chi-squared test [or the Fisher exact test] for categorical variables) indicated that a his-tory of steroid use ($p = 0.02$) and ASUC ($p = 0.001$) were risk factors for clinical relapse. Sex, age, duration of disease remission, extent of disease, and laboratory findings, such as white blood cell, hemoglobin, platelet, C-reactive protein, and albumin were not associated with clin-ical relapse. On multivariate analysis that included significant variables from the univariate analysis, ASUC was identified as an independent predictive factor for clinical relapse ($p = 0.024$, Odds Ratio- OR-: 21, 95% Confidence Interval- CI-: 1.50–293.2) (Table 3). Based

**Table 1. Baseline characteristics of the 70 patients.** 5-ASA: 5-aminosalicylic acids.

|  | No. or median | % or interquartile range |
|---|---|---|
| Total number of patients | 70 |  |
| Male gender | 43 | 61.4% |
| Age, years | 44 | 20–81 |
| Age at diagnosis, years | 34.5 | 17–78 |
| Duration of disease, years | 4.5 | 0–37 |
| Duration of disease remission, years | 1 | 0–10 |
| Extension of disease |  |  |
| Pancolitis | 28 | 40.0% |
| Left-side colitis | 19 | 27.1% |
| Proctitis | 23 | 38.6% |
| Partial mayo score | 0 | 0–1 |
| Type of oral 5-ASA |  |  |
| Pentasa | 43 | 61.4% |
| Asacol | 27 | 38.6% |
| White blood cell count, $10^3/\mu l$ | 5.6 | 3.7–14.2 |
| Hemoglobin, g/dl | 14 | 8.9–16.7 |
| Platelet, $10^3/\mu l$ | 225 | 115–553 |
| Albumin, g/dl | 4.4 | 3.7–5.2 |
| C-reactive protein $\geq$ 3mg/L | 5 | 7.1% |
| History |  |  |
| Thiopurines use | 3 | 4.3% |
| Corticosteroids use | 16 | 22.9% |
| Acute severe ulcerative colitis | 8 | 11.4% |

**Table 2. Univariate comparison between the patients who relapsed and those who maintained remission.**

|  | Remission | Relapse | Odds ratio | 95% CI | p value |
|---|---|---|---|---|---|
| Total number of patients | 52 (74.3%) | 18 (25.7%) |  |  |  |
| Male gender | 31 (44.3) | 12 (17.1%) | 0.74 | 0.24–2.28 | 0.78 |
| Age, years | 44 (21–81) | 44 (20–76) |  |  | 0.989 |
| Age at diagnosis, years | 35 (17–78) | 34.5 (21–66) |  |  | 0.682 |
| Duration of disease, years | 3 (0–37) | 6 (0–28) |  |  | 0.184 |
| Duration of disease remission, years | 1 (0–10) | 1 (0–9) |  |  | 0.594 |
| Extension of disease |  |  |  |  |  |
| Pancolitis | 23 (32.9%) | 5 (7.1%) | 0.41 | 0.11–1.49 | 0.207 |
| Left-side colitis | 14 (20.0%) | 5 (7.1%) | 0.67 | 0.18–2.54 | 0.739 |
| Proctitis | 15 (21.4%) | 8 (11.4%) | Reference |  |  |
| Partial mayo score:1 | 4 (5.7%) | 1 (1.4%) | 0.71 | 0.074–6.77 | 1 |
| Type of oral 5-ASA |  |  |  |  |  |
| Pentasa | 32 (45.7%) | 11 (15.7%) | 1.02 | 0.34–3.05 | 1 |
| Asacol | 20 (28.6%) | 7 (10.0%) | Reference |  |  |
| White blood cell count, $10^3/\mu l$ | 5.7 (3.7–11.3) | 5.6 (4.5–14.2) |  |  | 0.31 |
| Hemoglobin, g/dl | 13.95 (9.7–16.5) | 14.5 (8.9–16.7) |  |  | 0.377 |
| Platelet, $10^3/\mu l$ | 225 (115–553) | 216.5 (147–354) |  |  | 0.895 |
| Albumin, g/dl | 4.4 (3.7–5.2) | 4.5 (3.9–5) |  |  | 0.239 |
| C-reactive protein $\geq$ 3mg/L | 4 (5.7%) | 1 (1.4%) | 0.69 | 0.71–6.62 | 1 |
| History |  |  |  |  |  |
| Thiopurines use | 1 (1.4%) | 2 (2.9%) | 6.38 | 0.54–75.01 | 0.16 |
| Corticosteroids use | 8 (11.4%) | 8 (11.4%) | 4.4 | 1.33–14.55 | 0.02 |
| Acute severe ulcerative colitis | 2 (2.9%) | 7 (10.0%) | 15.91 | 2.90–87.23 | 0.001 |

on Kaplan–Meier survival analysis, the cumulative relapse-free survival rate within 52 weeks was 22.2% for patients with a history of ASUC, compared with 82.0% for those without. The log-rank test showed a statistically significant difference between patients with a history of ASUC and those without ($p < 0.001$) (Fig 2).

## Relapse after oral 5-ASA dose reduction

Patients were followed up for a median of 105 weeks (range: 5–298 weeks) after reducing the dose of 5-ASA (Fig 3). In total, 31 (44.3%) of the 70 patients relapsed at a median of 46 weeks (range: 5–266 weeks) after dose reduction. On Kaplan–Meier survival analysis, the estimated overall cumulative relapse-free survival rate was 74.3% at 52 weeks.

## Reinduction therapy

Of the 18 patients who experienced a relapse, 11 were treated with a high dose of oral 5-ASA, six with a high dose of oral 5-ASA and topical therapy, and one with oral corticosteroids as

**Table 3. Multivariable logistic regression analysis of independent variables associated with relapse during the 52 weeks after reducing the dose of oral 5-ASA.**

|  | Odds ratio | 95% CI | p value |
|---|---|---|---|
| History |  |  |  |
| Corticosteroids use | 0.733 | 0.08–6.79 | 0.785 |
| Acute severe ulcerative colitis | 21 | 1.50–293.2 | 0.024 |

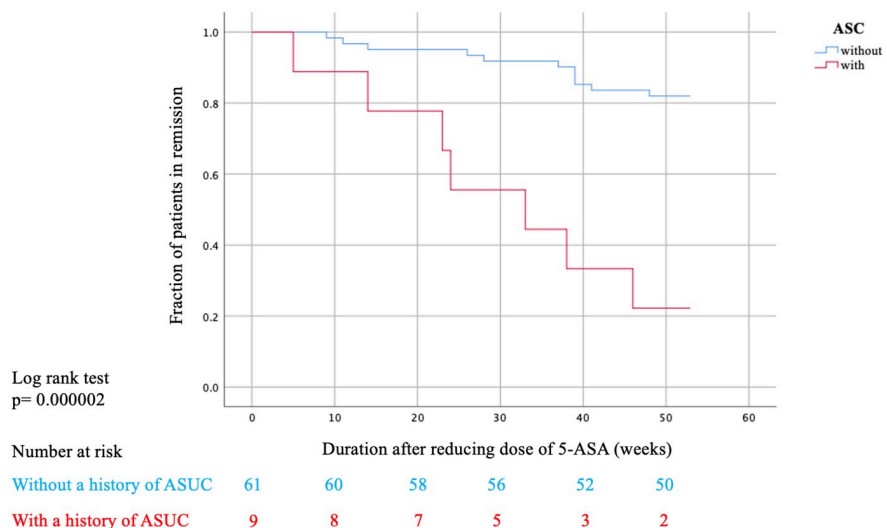

**Fig 2. Cumulative relapse-free survival rate with and without a history of ASUC.**

induction therapy. All patients subsequently achieved remission. None of the patients were hospitalized.

## Discussion

As shown in a meta-analysis, 5-ASA is safe and useful as maintenance therapy for UC [4], although there was insufficient evidence regarding the optimal dose of 5-ASA, and the risk factors for relapse after reducing the dose of 5-ASA were unknown. In the present study, 78% of patients with a history of ASUC relapsed at 52 weeks, compared with 18% of patients without a history of ASUC, indicating that patients with a history of ASUC are at a higher risk of relapse after a reduction in the dose of 5-ASA than those without a history of ASUC.

A total of 15% to 25% of patients with UC develop ASUC, which is a life-threatening condition, and require hospitalization [14, 17]. Very few studies to date have analyzed the long-term

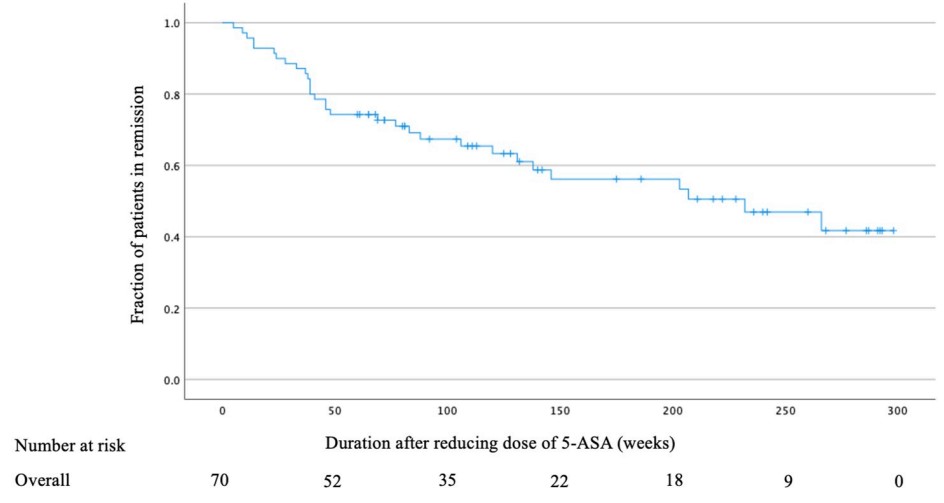

**Fig 3. Overall cumulative relapse-free survival rate after reducing the dose of oral 5-ASA.**

outcomes of patients with ASUC, particularly with respect to maintenance therapy [18, 19]. The ECCO recently considered that ASUC was an indicator of disabling disease, and that there was a need for accelerated step-up treatment strategies, i.e., thiopurine should be used for patients who are naïve to immunomodulators, and for patients who are refractory to thiopurine, and biological agents should then be used [5]. In this study, all patients with a history of ASUC were treated with intravenous steroids (IVS), whereas none of them were treated with anti-TNF agents or cyclosporine. The Toronto consensus recommends 5-ASA or immunomodulators as maintenance therapy for ASUC after induction therapy with IVS [20]. Several studies have reported the long-term prognosis of maintenance therapy for ASUC responding to IVS. Lotte et al. found that 39.8% (74/186) of patients experiencing one or more episodes of ASUC underwent a colectomy (OR: 11.81, 95% CI: 6.95–20.08, $p < 0.0001$) [14]. Salameh et al. reported relapse-free survival in 142 patients with ASUC on various maintenance therapy regimens who responded to IVS. After discharge from hospital, 90 (63.4%) of the patients experienced a relapse during their maintenance therapy, including 41 (69.5%), 37 (61.7%), and 9 (39.1%) patients who were treated with 5-ASA, immunomodulators, and biological agents, respectively. In addition, an early response to IVS and maintenance therapy with anti-TNF agents rather than 5-ASA or immunomodulators were associated with a lower rate of relapse [21].

Several studies reported that a history of steroid treatment was associated with a poor prognosis in UC patients [22–24]. Assessing the Safety and Clinical Efficacy of a New Dose of 5-ASA III trial, a multicenter, randomized, and double-blind study, assessed the efficacy and safety of treatment with 2.4 g/day and 4.8 g/day of mesalamine for 6 weeks in 772 patients with moderately active UC. A therapeutic advantage of 4.8 g/day was found in patients previously treated with corticosteroids, oral mesalamine, rectal therapy, or multiple UC medications. [24]. Fukuda et al. reported that the risk factors for clinical relapse in the low-dose mesalamine group were shorter duration to disease remission (p = 0.003, OR: 1.45, 95% CI: 1.13–1.89) and history of steroid use (p = 0.048, OR: 4.73, 95% CI: 1.01–22.2) [12]. Our results suggest that patients with a history of ASUC tend to have a higher rate of relapse than patients with a history of steroid use.

Duration of disease remission was not a risk factor for relapse. Although the overall recurrence rate of UC after 52 weeks was 25.7% in patients treated with oral 5-ASA in this study, the Cochrane review of 2016 reported that 41% of patients had experienced a relapse [4]. The reason for the low relapse rate in our study may be because many patients achieved long-term maintenance of remission or had high treatment adherence. Fukuda et al. emphasized that patients with frequent relapse should maintain remission with a high dose of oral 5-ASA [12]. In a prospective cohort study of 92 patients with UC during remission, Riley et al. reported that predictors of relapse were a shorter time from the previous relapse and a higher previous relapse rate [25]. The second reason is that the patients enrolled in this study, although including patients with ASUC, were not treated with anti-TNF agents or calcineurin inhibitors. In this regard, we considered that patients with refractory UC were not included in this analysis. Therefore, the relapse rate with dose reduction may not be so high.

Reducing the dose of 5-ASA may lead to an improvement of medication adherence and a reduction in medical expenses. Problems such as a high dose of 5-ASA and decreased adherence to medication may result in a higher rate of relapse [26, 27]. Khan et al. reported that there was no significant difference in the risk of relapse between a high and a low mesalamine dose in patients with high and moderate adherence [28]. The adherence rate for 5-ASA was found to be generally low in clinical practice, which likely resulted in the five-fold increase in the risk of recurrence, an increased risk of colorectal cancer, lower quality of life, and higher health care costs in both inpatient and outpatient setting [29, 30]. On the other hand, the

relapse rate may increase due to lower doses of 5-ASA. The annual cost per patient was estimated to be £1,693 for UC patients in remission, £2,903 for patients with mild to moderate UC, and £10,760 in patients with severe UC [31]. In addition to cost, relapses reduce patients' quality of life and productivity. 5-ASA has fewer side effects than other treatments for UC, so 5-ASA dose reductions should be used with caution.

The present study has some limitations. First, our study was a single-center, retrospective cohort study. Second, colonoscopy was not done in many of the patients, so the mayo endoscopic subscore (MES) was not included. There were 29 patients who received colonoscopy within 3 months prior to dose reduction of oral 5-ASA and 48 patients within 1 year (S1 Appendix). We did not determine criteria for reducing the dose of 5-ASA, and left it to the judgment of the doctor in charge. In patients in whom colonoscopy data were unavailable, attending physicians often reduced the dose of 5-ASA during clinical remission. Several studies have shown that endoscopic mucosal healing was associated with long-term disease remission. A post-hoc analysis of the Active Ulcerative Colitis Trials 1 and 2 trials found that patients who achieved an endoscopy score of 0 in week 8 had a four-fold increase in the likelihood of remission in week 30 [32, 33]. A prospective randomized controlled trial multicenter study should be performed in the future to confirm our results. In addition, as this study was conducted only on Japanese patients, patients of other races should also be analyzed in the future.

## Conclusion

Patients with a history of ASUC have an increased risk of relapse compared with patients without a history of ASUC. Therefore, 5-ASA doses should be reduced with caution in patients with a history of ASUC, as it is more likely to induce a relapse than in patients without a history of ASUC.

## Supporting information

**S1 Appendix.**
(XLSX)

## Acknowledgments

The authors would like to thank H. Popiel for the translation.

## Author Contributions

**Conceptualization:** Akira Madarame.

**Data curation:** Akira Madarame.

**Formal analysis:** Akira Madarame, Yoshiya Yamauchi.

**Investigation:** Akira Madarame, Yoshiya Yamauchi, Shin Kono, Akihiko Sugimoto, Hayato Yamaguchi, Takashi Morise, Yohei Koyama, Kumiko Uchida, Maya Suguro, Taisuke Matsumoto, Kagawa Yasuyuki.

**Methodology:** Akira Madarame, Masakatsu Fukuzawa, Takashi Kawai, Takao Itoi.

**Supervision:** Masakatsu Fukuzawa.

**Writing – original draft:** Akira Madarame.

**Writing – review & editing:** Masakatsu Fukuzawa.

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
