## [Decision Letter · Decision Letter 0]

7 Jun 2021

PONE-D-21-04470

Predictive factors of relapse after dose reduction of oral 5-aminosalicylic acid in patients with ulcerative colitis in the remission phase

PLOS ONE

Dear Dr. Madarame,

Thank you for submitting your manuscript to PLOS ONE. After careful consideration, we feel that it has merit but does not fully meet PLOS ONE’s publication criteria as it currently stands. Therefore, we invite you to submit a revised version of the manuscript that addresses the points raised during the review process.

We have revived the opinions of expert reviewers as agree with reviewers comments raised a few concerns about this study. We invite you to submit a revised version of the manuscript, please consider and address each of the comments raised by the reviewers.  

We look forward to receiving your revised manuscript.

Kind regards,

Senthilnathan Palaniyandi, Ph.D

Academic Editor

PLOS ONE

Journal Requirements:

2. In your ethics statement in the manuscript and in the online submission form, please ensure that you have discussed whether the IRB or ethics committee waived the requirement for informed consent. If patients provided informed written consent to have data/samples from their medical records used in research, please include this information.

4. Thank you for submitting the above manuscript to PLOS ONE. During our internal evaluation of the manuscript, we found significant text overlap between your submission and the following previously published works, some of which you are an author.

https://journals.plos.org/plosone/article?id=10.1371%2Fjournal.pone.0187737

Please revise the manuscript to rephrase the duplicated text, cite your sources, and provide details as to how the current manuscript advances on previous work. Please note that further consideration is dependent on the submission of a manuscript that addresses these concerns about the overlap in text with published work.

Reviewers' comments:

Reviewer's Responses to Questions

**Comments to the Author**

1. Is the manuscript technically sound, and do the data support the conclusions?

Reviewer #1: Yes

Reviewer #2: Yes

2. Has the statistical analysis been performed appropriately and rigorously? 

Reviewer #1: Yes

Reviewer #2: Yes

3. Have the authors made all data underlying the findings in their manuscript fully available?

Reviewer #1: Yes

Reviewer #2: Yes

4. Is the manuscript presented in an intelligible fashion and written in standard English?

Reviewer #1: Yes

Reviewer #2: Yes

5. Review Comments to the Author

Reviewer #1: This article is a study to determine the factors involved in the relapse in patients with ulcerative colitis in clinical remission when 5-ASA is reduced.

Although this is a fascinating retrospective study, there are several issues to be addressed.

1.Based on the current concept of T2T2, it is desirable to aim for mucosal healing (Mayo0) in maintaining long-term remission of ulcerative colitis. As the authors know, about 50% of patients present with endoscopic Mayo 1 or higher even when clinical remission is achieved. Thus, relapse may be related to endoscopic activity. Therefore, the endoscopic scores of the patients should have been shown in this study before starting the reduction of 5-ASA should be presented, although the author has described the limitation.

2. Patients enrolled in this study have not been treated with anti-TNF agents or calcineurin inhibitors, although patients with ASUC are included. In this regard, this reviewer considered that patients with refractory UC were not included in this analysis. Therefore, the relapse rate with dose reduction may not be so high. This reviewer speculates that patient-based factors may have contributed to no significant difference of duration of remission, etc., between relapser and non-relapser.

2 The authors should show fecal calprotectin before 5-ASA dose reduction.

3. The total amount of steroids used should be presented,

4. Exactly, the reduction of the 5-ASA dose can certainly reduce medical costs. However, the authors should consider the further increase in medical costs, decreased productivity, and decreased QOL of patients with UC if patients relapse after reducing 5-ASA. Therefore, the reduction of 5-ASA dose should be made with caution because 5-ASA has fewer side effects than other treatments. The authors should discuss the advantage and disadvantages of the 5-ASA reduction more and more.

Reviewer #2: The authors state that the patients who underwent combination therapy for UC, including corticosteroids, were excluded from the study. However, they also state that 22.9% of patients included in the study have used corticosteroids. Please explain.

Figures: Add Y-axes labels. Improve figure quality and appearance.

Typo: Last but one line: traial

6. PLOS authors have the option to publish the peer review history of their article (what does this mean?). If published, this will include your full peer review and any attached files.

Reviewer #1: No

Reviewer #2: No

---

## [Author Response · Author response to Decision Letter 0]

20 Jun 2021

We have carefully considered the points raised and made corrections. For more information, see Response to Reviewers.

---

## [Decision Letter · Decision Letter 1]

21 Jul 2021

Predictive factors of relapse after dose reduction of oral 5-aminosalicylic acid in patients with ulcerative colitis in the remission phase

PONE-D-21-04470R1

Dear Dr. Madarame,

We’re pleased to inform you that your manuscript has been judged scientifically suitable for publication and will be formally accepted for publication once it meets all outstanding technical requirements.

Kind regards,

Senthilnathan Palaniyandi, Ph.D

Academic Editor

PLOS ONE

Additional Editor Comments (optional):

Reviewers' comments:

Reviewer's Responses to Questions

**Comments to the Author**

1. If the authors have adequately addressed your comments raised in a previous round of review and you feel that this manuscript is now acceptable for publication, you may indicate that here to bypass the “Comments to the Author” section, enter your conflict of interest statement in the “Confidential to Editor” section, and submit your "Accept" recommendation.

Reviewer #1: All comments have been addressed

Reviewer #2: All comments have been addressed

2. Is the manuscript technically sound, and do the data support the conclusions?

Reviewer #1: Yes

Reviewer #2: Yes

3. Has the statistical analysis been performed appropriately and rigorously? 

Reviewer #1: Yes

Reviewer #2: Yes

4. Have the authors made all data underlying the findings in their manuscript fully available?

Reviewer #1: Yes

Reviewer #2: Yes

5. Is the manuscript presented in an intelligible fashion and written in standard English?

Reviewer #1: Yes

Reviewer #2: Yes

6. Review Comments to the Author

Reviewer #1: The author could not respond to all the comments of this reviewer. However, this reviewer judged that the author responded appropriately to the other contents. The quality of the English is somewhat poor, so this reviewer recommend that a native speaker check this manuscript again.

Reviewer #2: (No Response)

7. PLOS authors have the option to publish the peer review history of their article (what does this mean?). If published, this will include your full peer review and any attached files.

Reviewer #1: No

Reviewer #2: No

---

## [Editor Report · Acceptance letter]

26 Jul 2021

PONE-D-21-04470R1 

Predictive factors of relapse after dose reduction of oral 5-aminosalicylic acid in patients with ulcerative colitis in the remission phase 

Dear Dr. Madarame:

I'm pleased to inform you that your manuscript has been deemed suitable for publication in PLOS ONE. Congratulations! Your manuscript is now with our production department. 

Kind regards, 

on behalf of

Dr. Senthilnathan Palaniyandi 

Academic Editor

PLOS ONE